# Chitosan Nanoparticles Loaded with Quercetin and Valproic Acid: A Novel Approach for Enhancing Antioxidant Activity against Oxidative Stress in the SH-SY5Y Human Neuroblastoma Cell Line

**DOI:** 10.3390/biomedicines12020287

**Published:** 2024-01-26

**Authors:** Fadime Canbolat, Neslihan Demir, Ozlem Tonguc Yayıntas, Melek Pehlivan, Aslı Eldem, Tulay Kilicaslan Ayna, Mehmet Senel

**Affiliations:** 1Department of Pharmacy Services, Vocational School of Health Services, Çanakkale Onsekiz Mart University, Çanakkale 17100, Türkiye; 2Faculty of Science, Çanakkale Onsekiz Mart University, Çanakkale 17100, Türkiye; neslihandemir@comu.edu.tr; 3Faculty of Medicine, Çanakkale Onsekiz Mart University, Çanakkale 17100, Türkiye; oyayintas@comu.edu.tr; 4Vocational School of Health Services, İzmir Katip Çelebi University, İzmir 35620, Türkiye; pehlivanmlk@gmail.com; 5Medical Biology Department, Faculty of Medicine, İzmir Katip Çelebi University, İzmir 35620, Türkiye; aslierdem45@gmail.com (A.E.); tulayayna@gmail.com (T.K.A.); 6Tissue Typing Laboratory, İzmir Tepecik Education and Research Hospital, İzmir 35180, Türkiye; 7Department of Biochemistry, Faculty of Pharmacy, Biruni University, Istanbul 34010, Türkiye; msenel81@gmail.com; 8Department of Pharmaceutical Sciences, School of Pharmacy, University of California-Irvine, Irvine, CA 92697, USA

**Keywords:** antioxidant effect, chitosan, nanoparticle, oxidative stress, quercetin, valproic acid

## Abstract

Background: Multiple drug-delivery systems obtained by loading nanoparticles (NPs) with different drugs that have different physicochemical properties present a promising strategy to achieve synergistic effects between drugs or overcome undesired effects. This study aims to develop a new NP by loading quercetin (Que) and valproic acid (VPA) into chitosan. In this context, our study investigated the antioxidant activities of chitosan NPs loaded with single and dual drugs containing Que against oxidative stress. Method: The synthesis of chitosan NPs loaded with a single (Que or VPA) and dual drug (Que and VPA), the characterization of the NPs, the conducting of in vitro antioxidant activity studies, and the analysis of the cytotoxicity and antioxidant activity of the NPs in human neuroblastoma SH-SY5Y cell lines were performed. Result: The NP applications that protected cell viability to the greatest extent against H_2_O_2_-induced cell damage were, in order, 96 µg/mL of Que-loaded chitosan NP (77.30%, 48 h), 2 µg/mL of VPA-loaded chitosan NP (70.06%, 24 h), 96 µg/mL of blank chitosan NP (68.31%, 48 h), and 2 µg/mL of Que- and VPA-loaded chitosan NP (66.03%, 24 h). Conclusion: Our study establishes a successful paradigm for developing drug-loaded NPs with a uniform and homogeneous distribution of drugs into NPs. Chitosan NPs loaded with both single and dual drugs possessing antioxidant activity were successfully developed. The capability of chitosan NPs developed at the nanometer scale to sustain cell viability in SH-SY5Y cell lines implies the potential of intranasal administration of chitosan NPs for future studies, offering protective effects in central nervous system diseases.

## 1. Introduction

Valproic acid (2-n-propylpentanoic acid; VPA) is commonly prescribed for the treatment of migraine, bipolar disorder, epilepsy, and neuropathic pain [1,2]. In chronic use, the therapeutic concentration of VPA in plasma is typically 50–100 µg/mL for total VPA. However, VPA binds highly to albumin (approximately 95%), and as the plasma concentration increases, the binding capacity to albumin becomes saturated, leading to an increase in free (unbound) VPA levels. Free VPA, which is responsible for the pharmacological effects of the drug, can also contribute to toxicity [3,4,5]. Although various therapeutic ranges for free VPA plasma concentration have been suggested (e.g., 5–15 or 7–23 µg/mL), an optimal range has not been established [4]. While the use of VPA at low doses is generally considered safe, serious side effects such as hepatotoxicity, pancreatitis, and hyperammonemia encephalopathy can occur at high doses [1,2,5]. The exact causes of these toxicities related to VPA use have not been fully elucidated; however, disruptions in mitochondrial and fatty acid mechanisms arising from VPA biotransformation, reactive metabolites, carnitine deficiency, and the formation of oxidative stress are associated with VPA toxicity [1]. Various studies on the formation of oxidative stress due to VPA use in epilepsy patients have reported high levels of malondialdehyde, a biomarker of oxidative stress. It has been suggested that the formation of oxidative stress may contribute to side effects such as pancreatitis and liver toxicity [1]. Drugs that can help reduce oxidative stress damage often contain flavonoid structures that are responsible for antioxidant activity. One of the most commonly used molecules in this group with antioxidant properties is quercetin (Que). Que exhibits its antioxidant effect by scavenging reactive oxygen species (ROS) and chelating metal ions, thereby inhibiting the mechanism of lipid peroxidation [6]. The effective role of hydroxyl groups in the bioactivity of Que has been reported [6,7]. It is known that, in the long term, drugs used in chronic diseases can adversely affect many mechanisms in the body, especially oxidative stress. Bioactive molecules with high antioxidant activity, which are widely accepted as safe, are used to naturally reduce these adverse effects of drugs. Many studies highlight the potential protective effect of Que on drug toxicity. Canbolat et al. (2023) reported in an in vitro comet assay study that Que has a protective effect against genotoxic damage that may occur with VPA [8].

However, when evaluating the therapeutic efficacy of VPA and Que, various limitations arise in demonstrating the anticipated effects of these molecules. These limitations encompass VPA’s high binding to plasma proteins, the generation of toxic metabolites, severe side effects at elevated doses, restricted brain bioavailability of VPA due to challenges in crossing the blood–brain barrier (BBB) through anion transport proteins, and the potent antioxidant property of Que being impeded by rapid metabolism and low bioavailability. With current technological advancements, these constraints can be overcome by developing nanoparticle (NP) formulations. NP systems can offer numerous advantages in the field of drug delivery. Transporting drugs with NPs can enable more effective delivery to the target area, enhance bioavailability, and reduce the toxicity of the drugs [9]. NPs can increase the absorption and distribution of drugs in the body. NP systems can enhance drug absorption in the gastrointestinal system (GIS), improving the efficacy of drugs with traditionally low bioavailability. NP systems can reduce the first-pass metabolism of drugs in the liver, resulting in less metabolism and higher plasma concentrations of the drugs [10]. This, in turn, can enhance the systemic effects of the drugs. NPs, through the circulation drift effect in the bloodstream, allow drugs to circulate longer, increasing the chances of reaching the target area. NP systems can influence the half-life of drugs. The duration of a drug’s presence in the body can vary depending on the release rate and distribution properties determined by the NP system. NP systems can enable the controlled release of drugs. This allows for continuous and prolonged drug effects, reducing the frequency of dosages and minimizing the side effects. Transporting drugs through NPs can reduce toxicity by preventing the unwanted spread of drugs in the body. This contributes to making drug therapy safer in the field of health.

Furthermore, there are advancements in nanotechnological applications, including adding natural active substances to enhance antioxidant activity and cellular protection. Chitosan is significant in NP technology and is an FDA-approved, biologically degradable, mucoadhesive, biocompatible, and non-toxic polymer [11]. The median lethal dose (LD_50_) of chitosan is in the same range as sugar or table salt. Chitosan has three types of reactive functional groups: amino groups and primary and secondary hydroxyl groups. These groups allow for a variety of chemical modifications of chitosan [12]. There are a large number of advantages of chitosan for drug delivery, which include (i) the ability to control release; (ii) being free of toxic organic reagents while manufacturing NPs because they can dissolve in an acidic aqueous solution; (iii) a linear polyamine containing several free amine groups that are readily available for crosslinking; and (iv) cationic chitosan, which can be strongly combined with anionic material, cell surfaces, and mucous membranes, which benefits opening tight junctions, improving membrane absorption by prolonging the residual time at the site, and so on [13,14]. Compared with the polyester NPs usually used in brain delivery, chitosan NPs are unique for their positively charged surfaces and good mucosal adhesion. These characteristics could help enhance drug delivery to the brain through intranasal (i.n.) or intravenous routes (i.v.) [10,12]. Chitosan has demonstrated good antioxidant activity [6,7]. The literature reviews indicate the existence of characterization studies on chitosan–VPA NPs [15]. These studies show promising drug activities characterized by NPs using lower VPA doses. NP applications can provide the expected response from the drug while simultaneously reducing drug toxicity by preventing the formation of toxic metabolites. Both endogenous and exogenous molecular structures that can participate in the antioxidant system help alleviate the oxidative stress damage caused by drugs in organisms.

Multiple drug-delivery systems obtained by loading NPs with different drugs with different physicochemical properties present a promising strategy to achieve potential synergistic effects between drugs or overcome undesired effects that limit the benefits of many potential drugs or compounds. This study aims to develop a new NP by loading Que and VPA into chitosan. In this context, our study investigated the antioxidant activities of chitosan NPs loaded with single and dual drugs containing Que against oxidative stress. Accordingly, the synthesis of chitosan NPs loaded with a single drug (Que or VPA) and dual drugs (Que and VPA), the characterization of the synthesized NPs, the conducting of in vitro antioxidant activity studies, and the analysis of the cytotoxicity and antioxidant activity of the synthesized NPs on human neuroblastoma SH-SY5Y cell lines were performed to achieve these objectives.

## 2. Materials and Methods

### 2.1. Standard and Reagents

Chitosan (deacetylation degree 95%), quercetin dihydrate, sodium tripolyphosphate (STPP), valproic acid (VPA), 1-ethyl-3-(3-dimethylaminopropyl)-carbodiimide hydrochloride and N-hydroxysuccinimide, 1,1-diphenyl-2-picrylhydrazyl (DPPH), and 2,2-azinobis (3-ethylbenzothiazoline-6-sulfonic acid) diammonium salt (ABTS) were purchased from Sigma-Aldrich (Darmstadt, Germany).

### 2.2. Synthesis of Nanoparticles

The synthesis of the NPs in the study was carried out by modifying the methods of Bodnar et al., 2005, Zhang et al., 2008, Agarwal et al., 2018, Tzeyung et al., 2019, Wang et al., 2020, Jardim et al., 2022, and Messias de Souza et al., 2022 [7,16,17,18,19,20,21].

#### 2.2.1. Preparation of the Blank Chitosan Nanoparticle (NP1)

Blank chitosan NPs were synthesized by modifying the ionic chelation method in Bodnar et al., 2005, Zhang et al., 2008, Agarwal et al., 2018, Tzeyung et al., 2019, Jardim et al., 2022, and Messias de Souza et al., 2022 [7,16,17,18,19,21]. For a 1 mg/mL chitosan solution, 100 mg chitosan was dissolved in a 1.5% acetic acid solution. The pH of the solution was adjusted to 4.5 with 1 M sodium hydroxide solution. For a 1 mg/mL STPP solution, 100 mg STPP was dissolved in ultrapure water. The pH of the solution was adjusted to 4.5 through the addition of 10% hydrochloric acid. To 10 mL of chitosan solution (1 mg/mL pH: 4.5), 1 mg/mL STPP (pH: 4.5) was added dropwise for ionic chelation until the solution became turbid. The prepared sample was stirred at a low speed on a magnetic stirrer for 24 h.

#### 2.2.2. Preparation of the Quercetin-Loaded Chitosan Nanoparticle (NP2)

For the 3 mg/mL Que solution, 33.5 mg of quercetin dihydrate was weighed in a 10 mL flask and made up to volume with ethanol. The solution was kept in darkness. To 10 mL of chitosan solution (1 mg/mL pH: 4.5), 1 mg/mL STPP (pH: 4.5) was added dropwise for ionic chelation until the solution became turbid. A 1 mL volume of Que solution (3 mg/mL) was added to the mixture. The prepared sample was stirred at a low speed on a magnetic stirrer for 24 h.

#### 2.2.3. Preparation of Valproic Acid-Loaded Chitosan Nanoparticle (NP3)

The VPA and chitosan were conjugated by coupling carboxyl to amino group. Briefly, 2 mg/mL VPA was added to 10 mL of 1 mg/mL blank chitosan solution with 1 mL of 125 mM 1-ethyl-3-(3-dimethylaminopropyl)-carbodiimide hydrochloride and 1 mL of 125 mM N-hydroxysuccinimide. The prepared sample was mixed in a magnetic stirrer at a low speed for 24 h.

#### 2.2.4. Preparation of the Quercetin and Valproic Acid-Loaded Chitosan Nanoparticle (NP4)

A 2 mg/mL concentration of VPA and 1 mL of 3 mg/mL Que were added to 10 mL of 1 mg/mL blank chitosan solution with 1 mL of 125 mM 1-ethyl-3-(3-dimethyl aminopropyl)-carbodiimide hydrochloride and 1 mL of 125 mM N-hydroxysuccinimide. The prepared sample was mixed in a magnetic stirrer at a low speed for 24 h.

After 24 h of mixing, the samples were transferred to 50 mL falcon tubes and centrifuged at 8000 rpm for 40 min. After the centrifugation of all NPs, the filtrates were stored at −20 °C to determine the amount of free drug. The residues were lyophilized at −47 °C under 0.05 millibar for 24 h and used for characterization and antioxidant analyses. Samples were stored at −20 °C.

### 2.3. Characterization Analysis of the Nanoparticles

#### 2.3.1. Field Emission Scanning Electron Microscopy (FE-SEM) Analysis

The surface images were recorded using FE-SEM JEOL JSM-7100F a hot (Schottky) electron gun (JEOL, Pleasanton, CA, USA). To increase the conductivity properties of the samples, firstly, an 8 × 10^−1^ mbar/Pa vacuum was applied in the Quorum coating device (Quorum, East Sussex, UK), and a 10 mA voltage was applied to the gold–palladium (80–20%) coating process. Photographs were taken by applying a voltage of 10 kV. Analyses were run in three replicates.

#### 2.3.2. Transmission Electron Microscopy (TEM) Analysis

TEM analysis was performed using a JEOL JEM-1400 Plus 80 kV accelerating voltage (JEOL, Pleasanton, CA, USA). The samples were dispersed in water; then, a 10 µL solution was deposited on a carbon-supporting grid. After solvent evaporation, the TEM analyses were performed. Analyses were run in three replicates.

#### 2.3.3. Zeta Analysis

The zeta potential characterization of the nanoparticles and the determination of the polydispersity index (PDI) were carried out using a Zetasizer Nano ZS instrument (ZEN 3600- Malvern Panalytical, Worcestershire, UK) at a temperature of 25 °C. For the analysis, nanoparticles were dispersed in ultrapure water (approximately 0.01% by weight), sonicated for 10 min, and analyzed in triplicate at 25 °C.

#### 2.3.4. Fourier Transform Infrared Spectroscopy (FT-IR)

The samples’ infrared spectra were recorded using a Fourier Transform Infrared (FT-IR) Spectrum One (Perkin Elmer, Shelton, CT, USA) with the universal ATR sampling attachment (4000–450 cm^−1^) to examine the chemical structures. Analyses were run in three replicates.

#### 2.3.5. Encapsulation Efficiency

NP suspensions were centrifuged at 7000× *g* for 30 min. The free Que level in the supernatant at 415 nm (Que_λmax_) and the free VPA level at 302 nm (VPA_λmax_) were analyzed using an ultraviolet–visible (UV–Vis) spectrometer (Shimadzu Corporation, Kyoto, Japan). The encapsulation efficiency was determined using the efficiency calculation in Equation (1). Analyses were run in three replicates.
Encapsulation efficiency (%) = (T − F)/T × 100(1)

T: total amount of Que and VPA in solution (µg/mL);

F: amount of free Que and free VPA in the supernatant after centrifugation (µg/mL).

#### 2.3.6. Loading Capacity

NP suspensions were centrifuged at 7000× *g* for 30 min. The free Que level in the supernatant at 415 nm (Que_λmax_) and the free VPA level at 302 nm (VPA_λmax_) were analyzed using a UV–Vis spectrometer (Shimadzu Corporation, Kyoto, Japan). The loading capacity was determined using the calculation in Equation (2). Analyses were run in three replicates.
Loading capacity (%) = (T − F)/W × 100(2)

T: total amount of Que and VPA in solution (µg/mL);

F: amount of free Que and free VPA in the supernatant after centrifugation (µg/mL);

W: weight of dried NPs.

#### 2.3.7. In Vitro Release Analysis

The kinetics of Que and VPA release were analyzed by adapting the methodology described by Jardim et al. [21]. For this purpose, approximately 1 mg of NPs were suspended with phosphate-buffered saline (PBS) at pH 7.4. Samples were kept under constant stirring (700 rpm) in a thermostated water bath at 37 ± 0.1 °C. Each analysis was performed over 240 h (h), with measurements taken at predetermined intervals. Supernatants were analyzed in a UV–Vis spectrometer. Analyses were performed in three replicates.

### 2.4. Antioxidant Activity Analysis

The concentration range of the standard and the samples for antioxidant activity analysis was set at 0.05–0.6 mg/mL, and analyses were performed in three replicates. The DPPH radical scavenging activity of NPs was conducted following the method outlined by Blois et al. (1958) with slight modifications [22]. A 0.1 mM methanolic DPPH^•^ solution was added to NP suspensions, and the mixture was incubated in the dark for 30 min at room temperature. Absorbance values of the reaction solutions were recorded at 517 nm, and butylated hydroxytoluene (BHT) was used as the standard. DPPH^•^ radical scavenging activity was calculated using Equation (3).

The ABTS scavenging assay was performed as described by Re et al. (1999) [23]. ABTS^•+^ was generated via the oxidation of 7 mM ABTS with 2.45 mM potassium persulfate. Subsequently, the NP suspension was thoroughly mixed with the ABTS solution. After incubation in the dark for 15 min, the absorbance of the reaction mixture was recorded at 734 nm. The percentage scavenging of the ABTS^•+^ radical by the sample was calculated using Equation (3).
% scavenging activity= [(Abs_control_ − Abs_sample_)/Abs_control_] × 100(3)
where Abs_sample_ is the absorbance of the sample and Abs_control_ is the absorbance of the blank solution.

### 2.5. SH-SY5Y Cell Culture Conditions

The human neuroblastoma SH-SY5Y cell line was obtained from the American Type Culture Collection (CRL-2266™, ATCC, Manassas, VA, USA). The cells were cultured in a nutrient mixture consisting of Dulbecco’s modified Eagle’s medium (DMEM) (Gibco, Waltham, MA, USA) supplemented with 2 mM L-glutamine, 100 U/mL penicillin–streptomycin, and 10% heat-inactivated fetal bovine serum (FBS), which were all obtained from Gibco (Gibco, Waltham, MA, USA). The culture was maintained in a humidified atmosphere at 37 °C with 5% CO_2_ according to the protocol outlined in the literature [24].

### 2.6. Nanoparticle Treatments and Cytotoxic Evaluation

The cytotoxic effect of the samples was assessed using the MTT (3-[4,5-dimethylthiazol-2-yl]-2,5-diphenyltratrazolium bromide) (BioFroxx, Einhausen, Germany) assay. In a 96-well plate, SH-SY5Y cells were seeded at 5 × 10^3^ cells/well and allowed to attach to the surface for 24 h at 37 °C. The cells were exposed to 2 μg/mL, 8 µg/mL, 24 µg/mL, 48 µg/mL, 72 µg/mL, and 96 µg/mL concentrations of each sample for 24 h and 48 h at 37 °C and incubated for 4 h through the addition of 5 mg/mL of MTT. As the final step, after removing the MTT solution, each well was filled with 100 mL of dimethyl sulfoxide (DMSO), and the plate was stored at 37 °C for 10 min. Following the incubation period, cytotoxicity was determined spectrophotometrically at 570 nm using a Biotek™ Synergy™ HTX microplate reader (Santa Clara, CA, USA) [25]. The mean absorbance of the control cells was accepted as 100% viable. The inhibitory dose (IC_50_) value was calculated. The percentage cell viability was calculated using Equation (4).
% Cell viability = (Absorbance of cell treated cell)/(Absorbance of cell control cell) × 100(4)

### 2.7. Antioxidant Effect of Samples against Hydrogen-Peroxide-Induced Oxidative Stress in the SH-SY5Y Cell Line

The cytotoxic effect of H_2_O_2_ in SH-SY5Y cells was determined through MTT analysis [25]. For MTT analysis, SH-SY5Y cells were seeded in a 96-well plate at a density of 5000 cells per well in a 10% DMEM medium. The cells were then incubated in a 37 °C incubator with 5% CO_2_ for 24 h. After 24 h, the culture medium was removed from the wells, and the cells were exposed to 0 μg/mL, 2 μg/mL, 100 µg/mL, 200 µg/mL, 300 µg/mL, 500 µg/mL, and 1000 µg/mL concentrations of H_2_O_2_, which were then incubated for an additional 24 h. Each concentration of H_2_O_2_ was tested in four replicate wells. At the end of the incubation period, the culture medium containing H_2_O_2_ was discarded, and 5 mg/mL of the MTT solution was added to each well. The cells were further incubated for 4 h. After incubation, the medium containing insoluble formazan crystals was removed, and the crystals were dissolved in 100 µL of DMSO. Absorbance at 570 nm was measured using a spectrophotometer. The viability of the H_2_O_2_ group was compared to the control group, where cell viability was considered 100%. The H_2_O_2_ dose that reduced cell viability by 50% (IC_50_; 200 µM) was determined, and oxidative stress was induced in SH-SY5Y cells. Subsequently, antioxidant activity analyses of NPs in SH-SY5Y cells were conducted. To evaluate the antioxidant effects of NPs, SH-SY5Y cells were exposed to 200 µM H_2_O_2_. The cells were initially seeded at a density of 5 × 10^3^ cells/well in a 96-well plate and cultured overnight. Subsequently, 10 μL of NPs at various concentrations (2 μg/mL, 8 μg/mL, 24 μg/mL, 48 μg/mL, 72 μg/mL, and 96 μg/mL) was administered at 24 h and 48 h before treatment with 200 µM H_2_O_2_.

### 2.8. Statistical Evaluation

Logarithmic calculations were made from the dose–response graph GraphPad Prism 9 to obtain the IC_50_ values. Data are presented as the mean of triplicate experiments ± SD (****, *p* < 0.0001; ***, *p* < 0.0005; **, *p* < 0.01; *, *p* < 0.05; ^####^, *p* < 0.0001; ^###^, *p* < 0.0005; ^##^, *p* < 0.01; ^#^, *p* < 0.05). Results are presented as mean (M) ± standard deviation (SD) of three replicate experiments. A one-way ANOVA test was used to evaluate the differences between groups in antioxidant assays, and the post hoc Tukey HSD test was used to compare groups.

## 3. Results

The FE-SEM and TEM images of the NPs developed by the ionic chelation method shown in Figure 1 and Figure 2 reveal that NP1 and NP3 are spherical in shape, while NP2 and NP4 have a strip-like structure.

Table 1 shows the NPs’ particle size (nm) and zeta potential (mV) values according to TEM (Figure 2) and zeta analysis (Figure 3) results. The mean particle size and zeta potential values of NP1, NP2, NP3, and NP4 were 153.6 ± 12.81 nm and 16.6 ± 4.12 mV, 180.5 ± 8.23 nm and 15.1 ± 3.42 mV, and 93.2 ± 7.25 and 8.89 ± 3.82, 198.7 ± 9.13 and 27.2 ± 3.67, respectively. According to the results of the zeta analysis, the sharpness of the chromatograms, the zeta potential, and low PDI values indicate a stable and homogeneous distribution of the developed NPs within the solution. However, the stability of NP3 appears to be lower than the other NPs (Table 1, Figure 3).

The encapsulation efficiency of Que and VPA in the NPs is above 95%, as calculated in Equation (1) (Table 1). Table 1 demonstrates a high encapsulation efficiency of Que and VPA loaded into chitosan. The drug-loading capacities of the drugs loaded into chitosan in the synthesized chitosan NPs shown in Table 1 reveal that Que was present in the NPs at a rate of approximately 62.22% and 39–16% in NP2 and Np4, respectively. VPA was found in NPs in NP3 and NP4 at approximately 37.71% and 25.75%, respectively. The dual-drug-loading rate in NP4 was approximately 64.91%. While high drug-encapsulation efficiency was achieved in the synthesized NPs, medium drug-loading capacity was obtained.

The spectrum of samples is shown in Figure 4 The broad peak observed at 2884 cm^−1^ in the FT-IR spectrum of chitosan is due to the C-H bonds’ symmetric and asymmetric stretching vibrations (Figure 4). The band at 1653 cm^−1^ (C=O stretching in the amide group, amide I vibration) was seen in chitosan [18,26]. The peak at 1545 cm^−1^ is ascribed to the N−H bending of the amino group [27]. The peak at 1027 cm^−1^ is due to the C−O stretching of the glucosamine ring. Similar results were obtained in the FT-IR spectrum of the synthesized NP1.

In Que’s FT-IR spectrum, the broad peak observed at 3414 cm^−1^ corresponds to O-H stretching vibration, while the peak observed at 1666 cm^−1^ corresponds to C=O aryl ketonic stretching vibration (Figure 4). The 1518 cm^−1^ and 1613 cm^−1^ peaks correspond to C=C aromatic ring stretching vibrations. The peak at 1263 cm^−1^ corresponds to the C-O stretching vibration in the aryl ether ring. Although similar results were obtained in the FT-IR spectrum of NP2, the peak at 1324 cm^−1^ appears to correspond to the C-H stretching vibration of the aromatic ring [28]. When Figure 4 is examined, it shows that the C=O stretching vibration observed at 1664 cm^−1^ confirms the presence of Que in chitosan.

Upon examination of Figure 4, it is evident that both the VPA standard and the NPs containing VPA (NP3 and NP4) exhibit similar peaks at approximately 2961 cm^−1^ and 1700.6 cm^−1^. In the FT-IR spectrum of VPA, the broad peaks observed at 2961.3 cm^−1^ indicate aliphatic O-H stretching vibration, while the peak at 1700.6 cm^−1^ corresponds to C=O stretching vibration [29,30]. Additionally, peaks attributed to Que, specifically the C=C aromatic ring, are observed at 1612.3 cm^−1^ and 1521.3 cm^−1^ in NP4 (Figure 4). This observation reveals that NPs can be appropriately developed.

Our examination of the release of NPs at pH 7.4 and 37 ± 0.1 °C indicated that Que exhibited a rapid release within the first 10 h. In contrast, the burst release of VPA from chitosan NPs containing VPA commenced at the 50th hour in NP3 and at an earlier time, the 30th hour, in NP4 (Figure 5).

DPPH^•^ and ABTS^•+^ radical scavenging activities are frequently used in studies investigating the antioxidant effect of Que-loaded NPs. In our study, the antioxidant activity of NPs was determined using these methods. Each NP’s antioxidant activity within the dose range of 0.05–0.6 mg/mL was compared with the standard antioxidant activity of BHT. The analysis data are depicted in Figure 6. In the DPPH and ABTS analysis results, the antioxidant activity of NP1 and NP3, which do not contain Que, was found to be lower than that of BHT at all concentrations. A statistically significant difference was found between BHT, NP1, and NP3 antioxidant activity results in both antioxidant activity analyses (*p* < 0.05). In the results of DPPH analysis, the antioxidant activity of NP2 and NP4, which contain Que, was comparable to or higher than the antioxidant activity of BHT. Antioxidant activity increased dose-dependently. The antioxidant activity of BHT varied between 49.77% and 90.84%, while the antioxidant activity of NP2 appeared to range from 74.67% to 88.33%, and NP4’s antioxidant activity was observed to be between 63.19% and 90.15%.

In DPPH analysis, the antioxidant activity of NP2 was higher than that of BHT at the 0.05, 0.1, and 0.2 mg/mL levels, while it was close to the activity of BHT at the 0.4 and 0.6 mg/mL levels. The antioxidant activity of NP4 was higher than that of BHT at the 0.05 and 0.1 mg/mL levels, while it was close to the activity of BHT at the 0.2, 0.4, and 0.6 mg/mL levels. NP2 showed the highest DPPH-scavenging activity at 0.4 mg/mL, and NP4 showed the highest DPPH-scavenging activity at 0.6 mg/mL (Figure 6A).

In the ABTS analysis results, although the antioxidant activity of NP2 and NP4 was lower than that of BHT, a dose-dependent antioxidant activity was observed, similar to the BHT standard regarding the antioxidant effect. NP2 and NP4 showed the highest ABTS^•+^ radical scavenging activity at a concentration of 0.6 mg/mL (83.21 ± 4.91 and 77.70 ± 7.27, respectively). The antioxidant analysis data showed that NP2 and NP4 containing Que exhibited antioxidant activity close to BHT, while NP1 and NP3 were the NPs with the lowest antioxidant activity (Figure 6B).

To evaluate the cytotoxicity and antioxidant effect of the synthesized NPs on human neuroblastoma SH-SY5Y cell lines, the H_2_O_2_ value to be applied to the cells was determined. As a result of the MTT test, IC_50_ value was determined as 200 µM (Figure 7).

Que- and VPA-loaded chitosan NPs were applied to SH-SY5Y cells at different doses, and cell cytotoxicity was evaluated. Except for high doses of NP4, none of the samples showed cytotoxic effects at the concentrations used in this study. At the concentration of 96 µg/mL applied to the cells, NP4 was observed to reduce cell viability by approximately 50% at 24 h (Figure 8) and by around 40% at 48 h (Figure 9).

To assess the antioxidant potential of NPs against oxidative stress, SH-SY5Y cells were subjected to pretreatment with different concentrations of NPs for 24 h and 48 h, followed by exposure to 200 µM H_2_O_2_ for 24 h. While 50% cell viability was observed in SH-SY5Y cells treated solely with H_2_O_2_, cell viability increased when cells were pretreated with NPs before exposure to H_2_O_2_. Additionally, it was observed that when VPA, Que, and chitosan molecules present in the composition of NPs were applied to SH-SY5Y cells exposed to 200 µM H_2_O_2_ at different concentrations (2 µg/mL, 8 µg/mL, 24 µg/mL, 48 µg/mL, 72 µg/mL, and 96 µg/mL), these molecules exhibited antioxidant activity, thereby exerting a protective effect on cell viability (Figure 10 and Figure 11).

When 48 µg/mL, 72 µg/mL, and 96 µg/mL of VPA were applied to the cells, it was determined that cell viability increased at 24 h and 48 h. Cell viability was significantly elevated with the increase in 48 µg/mL (*p* < 0.01), 72 µg/mL (*p* < 0.01, *p* < 0.05), and 96 µg/mL (*p* < 0.0001, *p* < 0.01) at 24 h and 48 h, respectively. At 24 h, the increase in the dose resulted in a cell viability range of 52.77% to 57.89% (Figure 10), while at 48 h, the increase in the dose led to a cell viability range of 52.7% to 55.83% (*p* < 0.01) (Figure 11). The highest efficacy was achieved at 24 h through the application of 96 µg/mL VPA, resulting in 57.89% cell viability (*p* < 0.0001). However, it was observed that low and moderate doses of VPA (2 µg/mL, 8 µg/mL, and 24 µg/mL) did not have a significant impact on cell viability (*p* > 0.05) (Figure 10).

When Que was applied to the cells at different doses during both intervals, the lowest dose did not show a noticeable effect on cell viability. At 24 h, the cell viability was 49.95% in cells treated with 2 µg/mL of Que, and this rate decreased to 47.06% at 48 h (*p* < 0.05), resulting in a significant difference compared with the control group. However, the application of 8 µg/mL Que began to demonstrate efficacy by increasing cell viability by 59.8% at 24 h (*p* < 0.0001) (Figure 10). At the same time, no significant effect was observed at this dose at 48 h (Figure 11). It was observed that the application of 24 µg/mL, 48 µg/mL, 72 µg/mL, and 96 µg/mL Que significantly increased cell viability with dose increment at both 24 h and 48 h. At 24 h, cell viability increased by 61.3% to 66.0% with the dose increment, while at 48 h, cell viability increased by 58.32% to 65.99% with the dose increment (*p* < 0.0001). The highest efficacy was achieved by applying 96 µg/mL Que at 24 h, resulting in 66.0% cell viability.

When chitosan was applied to the cells at different doses, it was observed that low doses of chitosan (2 µg/mL and 8 µg/mL) did not significantly increase cell viability at 24 h (*p* > 0.05) (Figure 10). However, at 48 h, these low doses significantly increased cell viability, reaching approximately 60% (*p* < 0.0001) (Figure 11). It was determined that the application of 24 µg/mL, 48 µg/mL, 72 µg/mL, and 96 µg/mL of chitosan significantly increased the cell viability, with dose escalation at both 24 h and 48 h. At 24 h, cell viability increased in the range of 52.54% to 61.04% with dose escalation, while at 48 h, cell viability increased in the range of 62.12% to 66.31% with dose escalation (*p* < 0.0001). The highest efficacy was achieved by applying 96 µg/mL chitosan at 48 h (*p* < 0.0001).

When NP1 was applied to the cells at different doses, it was observed that the low and moderate doses of NP1 (2 µg/mL, 8 µg/mL, 24 µg/mL, and 48 µg/mL) did not significantly increase cell viability at 24 h (*p* > 0.05) (Figure 10). However, at 48 h, these doses were found to significantly increase cell viability (*p* < 0.01, *p* < 0.05) (Figure 11). NP1 treatment exhibited a protective effect at concentrations of 72 µg/mL and 96 µg/mL against H_2_O_2_ toxicity at both time points (24 h and 48 h) (*p* < 0.0001). It was determined that applying 72 µg/mL NP1 at 24 h and 48 h increased cell viability to approximately 60% (*p* < 0.0001). Additionally, the application of 96 µg/mL NP1 at 24 h and 48 h elevated cell viability to above 65%, with the highest efficacy observed at 48 h with 96 µg/mL NP1, reaching 68% cell viability (*p* < 0.0001).

Upon application of NP2 at different doses to the cells, it was observed that, at 24 h, only the low dose (2 µg/mL) did not significantly increase cell viability (*p* > 0.05) (Figure 10). However, at 48 h, it was observed that low and moderate doses (2 µg/mL, 8 µg/mL, and 24 µg/mL) did not significantly increase cell viability (*p* > 0.05) (Figure 11). When NP2 levels of 8 µg/mL, 24 µg/mL, 48 µg/mL, 72 µg/mL, and 96 µg/mL were applied to the cells for 24 h, a significant increase in cell viability of 55.49% to 66.53% was observed (*p* < 0.0001). The highest efficacy was achieved by applying 48 µg/mL NP2 at 24 h (*p* < 0.0001). However, when NP2 levels of 48 µg/mL, 72 µg/mL, and 96 µg/mL were applied to the cells for 48 h, significant increases in cell viability of 60.38% to 77.30% were observed (*p* < 0.0001). NP2 demonstrated a protective effect against H_2_O_2_-induced cytotoxicity at all doses, with a more pronounced effect at higher doses. Particularly, at 96 µg/mL, it exhibited a 77.30% protective effect against H_2_O_2_-induced cytotoxicity.

In NP3 applications, cell viability remained above 60% for all dose applications at both time points (Figure 10 and Figure 11). However, in the 24 h cell viability analysis, the cell viability ratio was significantly high for all doses (60.24–70.06%), and as the dose increased, the increase in cell viability decreased (Figure 10). As shown in Figure 10, NP3 demonstrated the highest antioxidant effect when applied at 2 μg/mL, maintaining cell viability up to 70%.

In the 24 h application, it was observed that the low and moderate doses of NP4 (2 µg/mL, 8 µg/mL, 24 µg/mL, and 48 µg/mL) significantly increased cell viability up to 66% (Figure 10). In the 24 h pretreatment experiment, NP4 at 72 µg/mL and 96 µg/mL provided lower protection against neurotoxicity (Figure 10). In contrast, 2 and 8 µg/mL NP4 preserved cell viability up to 66.03% and 63.18% against H_2_O_2_, respectively. NP4 exhibited the highest viability at a concentration of 2 µg/mL at 24 h. In other words, the dose with the best protective effect for NP4 was 2 µg/mL at 24 h. In the 48 h pretreatment experiment, the low, moderate, and high doses (2 µg/mL, 8 µg/mL, 24 µg/mL, 48 µg/mL, 72 µg/mL, and 96 µg/mL) were observed to significantly increase cell viability in the range of 54.13% to 64.09% for all doses (Figure 11). The efficiency of 48 µg/mL, 72 µg/mL, and 96 µg/mL NP4 was lower, whereas 2 µg/mL, 8 µg/mL, and 24 µg/mL NP4 protected cell viability up to 60.78%, 60.63%, and 64.09% against H_2_O_2_, respectively.

In cell culture analyses, it was observed that all NPs increased cell viability against H_2_O_2_-induced cell damage by using a 10× magnification fluorescence microscope Olympus CK40M (Olympus Corporation, Tokyo, Japan) (Figure 12).

## 4. Discussion

The synthesis of the NPs in this study was carried out by modifying the methods of Bodnar et al., 2005, Zhang et al., 2008, Agarwal et al., 2018, Tzeyung et al., 2019, Wang et al. 2020, Jardim et al., 2022, and Messias de Souza et al., 2022 [7,16,17,18,19,20,21]. The preparation of chitosan NPs was based on an ionic gelation interaction between positively charged chitosan and negatively charged STPP at room temperature. The chitosan NPs prepared in this experiment were in white powder form, similar to how they were in the literature [7,16,17,18,19,20,21]. The Que stock solution was prepared in ethanol, as in the literature [7,17,21]. NP1 and NP3 were white, while NP2 and NP4 were yellowish, indicating that the Que was loaded into the chitosan. In synthesizing NPs, pH, ultrasonic bath soaking time, stirring speed, and duration were considered. In our study, when FE-SEM (Figure 1) and TEM (Figure 2) images were examined, it was seen that NP1 and NP3 were nearly spherical in shape [31], and NP2 and NP4 were strip-like structures [32]. The FE-SEM and TEM images of the NPs synthesized in our study are similar to those of NPs containing Que and VPA in the literature [32,33,34,35,36]. Images and zeta potential values of NPs kept in an ultrasonic bath at pH 4.5 for 5 min and stirring at low speed for 24 h are given in Table 1 and Figure 3.

The mean particle size and zeta potential value of NP1 and NP2 were 153.6 ± 12.81 nm and 16.6 ± 4.12 mV and 180.5 ± 8.23 nm and 15.1 ± 3.42 mV, respectively (Table 1). The positive zeta potential of NP substrate assisted the suitable interaction and ionic crosslinking of NPs to negatively charged molecules. This clearly shows that the loading of Que in chitosan NPs causes a significant decrease in the zeta potential. A similar situation was also reported in Roy and Rhim’s study (2021) [6]. Li et al., 2018, performed single- and double-molecule loadings to chitosan. In the study, Que loaded on chitosan NPs’ particle size was 190.7 ± 2.8 nm [36]. The particle size of the NPs obtained in our study is similar to the literature.

In our study, NP1 and NP3 were formulated with a mean diameter of 153.6 ± 12.81 nm and 93.2 ± 7.25 nm for chitosan and VPA-loaded chitosan samples, respectively (Table 1). The loading of VPA into chitosan led to a decrease in size by about 60 nm. The results may be attributed to the electrostatic interaction and ionic crosslinking between negatively charged VPA and positively charged chitosan molecules, which led to electric attraction force among substrates and final shrinkage of the particles. A similar situation to our study was also observed in the study conducted by Jafarimanesh et al. [33]. In their 2023 study, Jafarimanesh et al. developed chitosan NPs loaded with VPA. When the particle sizes of the synthesized NPs in Jafarimanesh et al.’s study were examined, a decrease in particle size was observed with VPA loading. In our study, in NP4, where dual drug loading occurred, the NP size increased with the addition of Que. Therefore, the data from our study demonstrate similarity with the literature. The biochemical influence of scaffolds was a key factor in their applicability, and great attention was required for the particle surface characteristics and particle size distribution. The magnitude of the zeta potential provides information about particle stability [33]. To evaluate the stability of the synthesized NPs, zeta potential measurements were used. Higher magnitude potentials indicate increased electrostatic repulsion and, thus, increased stability. Particles with zeta potentials in the 0–5 mV range tend to aggregate or cluster, particles in the 5–20 mV range are stable, and particles above 20 mV are highly stable [37]. The zeta potential data in Table 1 and Figure 3 show that the zeta potential values of NP1, NP2, and NP3 were in the range of 5–20 mV and showed a stable structure. The NP with the highest zeta potential and exhibiting high stability in our study was identified as NP4. The proximity of the PDI values of all developed NPs to zero in our study indicates a homogeneous nanoparticle distribution (Table 1). This observation suggests that each NP exhibited similar sizes, and their distributions were not widely spread. In addition, the sharpness of the zeta potential peak in Figure 3 provides information about the homogeneity of the structure. When examining the results of the zeta analysis, the sharpness of the chromatograms and the potential values indicate a stable and homogeneous distribution of the developed NPs within the solution. All obtained results indicate that drugs were homogeneously loaded into NPs, and this loading was uniformly distributed. This situation may support the effectiveness of drug transport, enabling the drug to reach the targeted area more efficiently. Thus, it suggests a potential contribution to enhancing the bioavailability of the drugs.

Encapsulation efficiency is a crucial factor in drug-delivery release. For the sustained release of an adequate amount of the drug at the target site, a high level of entrapment of the drug into the NPs is expected [21,38]. The encapsulation efficiencies of Que NPs developed with different methods in the literature were obtained as 79.8 ± 1.5% [39], 83.8 ± 0.33% [21], and 92% [40]. In our study, when the amounts of drugs in the filtrates were analyzed for encapsulation efficiency, it was observed that due to the trace amounts of drugs in the filtrates, drug loading in the NPs was achieved with high encapsulation efficiency. In our study, the Que-encapsulation efficiency was 93.46 ± 5.52% and 99.23 ± 4.73% in NP2 and NP4, respectively, while the VPA-encapsulation efficiency was 99.9 ± 7.88% and 97.8 ± 5.42% in NP3 and NP4, respectively (Table 1). In our study, NPs with higher encapsulation efficiency were synthesized, compared with the literature. This is because adjusting the optimal solvent, pH, stirring speed, and time during synthesis effectively developed NPs. The drug-loaded and encapsulation rates were very important criteria for judging the drug-loaded performance. Table 1 shows that when quercetin is singly loaded into chitosan, the drug-loading capacity is approximately 62.22%. In contrast, for chitosan loaded with dual drugs, the drug-loading capacity for quercetin is approximately 39.16%. Similarly, when VPA is singly loaded into chitosan, the drug-loading capacity is approximately 37.71%, whereas for chitosan loaded with dual drugs, the drug-loading capacity for VPA is approximately 25.75%. The total drug-loading capacity of chitosan NPs loaded with dual drugs is 64.91%. NPs with high encapsulation efficiency and moderate drug-loading capacities have been synthesized using the developed analysis method. However, considering the high encapsulation efficiency of quercetin and VPA in the NPs and taking into account the drug-unloaded regions in chitosan, it is believed that with minor modifications in the developed synthesis method (such as increasing the drug amount and the mixing and centrifugation times), similar encapsulation efficiency can be achieved, resulting in higher drug-loading capacities. Consequently, administering these nanoparticles to the body system is anticipated to yield a higher efficiency than the drug alone.

FT-IR characterization reveals the intermolecular interaction of chitosan NPs. IR spectroscopy is an extremely effective method for determining the presence or absence of various functional groups in a molecule. According to the results of FT-IR analysis in the chitosan spectrum, the bands 1653 cm^−1^ (C=O stretching in amide group, amide I vibration) and 1546.4 cm^−1^ (N-H bending), respectively, in chitosan, shift to 1636 cm^−1^ and 1518.4 cm^−1^ for NP1 and 1664.2 cm^−1^ and 1524 cm^−1^ for NP2 due to the interaction between the phosphoric groups of STPP and the amino groups of chitosan in NPs. Thus, it is postulated that the polyphosphoric groups of STPP interact with the ammonium groups of chitosan, enhancing both the inter- and intramolecular interaction in chitosan NPs [18]. Similar results were observed by Agarwal et al. [18]. FT-IR spectroscopy revealed that both spectra of Que standard and NP2 have the same characteristic peaks of O-H (3400 cm^−1^) and aromatic C=C (1510 cm^−1^, 1610 cm^−1^). Still, the intensity of the sample peaks is lower (Figure 4). The peaks at 3414 cm^−1^, 1613.7 cm^−1^, and 1518.4 cm^−1^ appearing in the FT-IR analysis of the Que standard are observed at approximate values in NP2. In Roy and Rhim’s study, 1614 cm^−1^ was noted as the prominent peak of Que [6]. This observation reveals that NPs can be developed appropriately, as our data are similar to those from other studies in the literature [6,24,26,28]. Upon examination of Figure 4, it is evident that both the VPA standard and the NPs containing VPA (NP3 and NP4) exhibit similar peaks at approximately 2961 cm^−1^ and 1700.6 cm^−1^. In the FT-IR spectrum of VPA, the broad peaks observed at 2961.3 cm^−1^ indicate aliphatic O-H stretching vibration, while the peak at 1700.6 cm^−1^ corresponds to C=O stretching vibration [29,30]. Additionally, peaks attributed to Que, specifically the C=C aromatic ring, are observed at 1621 cm^−1^ and 1521 cm^−1^ in NP3 and NP4. This observation reveals that nanoparticles can be appropriately developed.

The in vitro release studies were performed at pH 7.4, which was equivalent to that of brain pH [15]. The results show that the VPA and Que release profiles from chitosan NPs were rapid in our study (Figure 5). In the study by Jaiswal et al., they prepared the repurposed VPA-loaded polymeric nanoparticles using carboxymethyl chitosan for the management of Alzheimer’s disease through the intranasal route. The drug release was observed for 48 h in a buffer solution at pH 7.4, and the drug release was found to be 82% [15]. In the study by Jafarimanesh et al. in 2023, VPA was loaded into chitosan NPs within a hybrid of alginate/chitosan hydrogel. The study observed a rapid release of VPA from the chitosan hydrogel loaded with VPA, with the initial burst release of VPA observed after 72 h of incubation [33]. In our study, half of the VPA loaded into chitosan NPs was released at 40 h in NP3 and 24 h in NP4. The burst release of VPA from chitosan NPs containing VPA started at 50 h in NP3 and at an earlier time, 30 h, in NP4 (Figure 5A). Raj et al., 2015 reported that the release of Que was rapid in the first 10 h [35]. In our study, when the release of NPs at pH 7.4 37 ± 0.1 °C was examined, it was observed that Que was released rapidly within the first 8 h (Figure 5B). The data in our study are similar to those in the literature. The differential release profiles between NP3 and NP4 for VPA and the rapid release of Que within the first 8 h at pH 7.4 37 ± 0.1 °C are important findings in our study. For the NPs with high drug-encapsulation efficiency in our study, we can primarily consider two scenarios based on the potential drug effects when these nanoparticles enter the body system. The first scenario involves individuals who may have oxidative stress damage due to any disease or xenobiotic exposure, and the second scenario pertains to the effects that may occur in patients undergoing VPA treatment. In both scenarios, it is anticipated that individuals using NPs containing Que (NP2) will experience the rapid and controlled release of Que from the NP in the body system. Consequently, the pharmacological effects of Que against oxidative stress are expected to be observed quickly. However, in the second scenario, individuals using NP4 as a dual drug system (Que and VPA loaded on chitosan) will witness the release of Que into the body system before VPA. With its reparative effect against oxidative stress, Que will begin to demonstrate its effect on the environment beforehand. Due to the slower and controlled release of VPA compared with Que, avoiding sudden peaks in VPA dosage is expected. This not only helps avoid toxicity associated with VPA dosage but also allows for the utilization of the reparative effect of VPA against oxidative stress within its safe dose range. In future studies, the facilitated targeting of advanced NPs to specific regions through different administration routes (e.g., intranasal) could enhance the ease of reaching target areas. This could potentially lead to the reliable observation of the antiepileptic treatment effect of VPA at lower doses.

Due to the antioxidant properties of Que, Que-loaded NPs have been developed in the literature using different nanostructures. The antioxidant activities of NPs reported in the literature include Li et al., 2018, who reported that genipin and Que-loaded chitosan NPs showed antioxidant activity in DPPH and ABTS assays [36]. Wu et al. (2008) reported that NPs composed of Que, polyvinyl alcohol, and aminoalkyl methacrylate copolymers showed antioxidant activity in vitro [41]. Zhang et al. (2008) studied the radical removal activity of Que-loaded chitosan NPs in the 0.5–3 mg/mL range. As a result of the analysis, it was reported that the NPs showed antioxidant activity [17]. Unlike Zhang et al.’s (2008) study, our study investigated the radical scavenging activity and antioxidant capacity of NPs at lower concentrations. In our study, antioxidant activity analysis of samples in the range of 0.05–0.6 mg/mL was performed. In our study, determining the antioxidant capacities of the NPs developed against ROS was among our research objectives. To achieve this, in vitro antioxidant capacity analyses were conducted to determine the effectiveness of the NPs. In our study, the antioxidant activity capacity of NPs containing Que was observed to be close to the antioxidant activity capacity of the BHT standard (Figure 6). Effective antioxidant activity was observed in NP2 and NP4. However, NP1 and NP3 without Que showed significantly lower antioxidant activity than the BHT standard (Figure 6). In a study conducted by Palol et al. (2021), the antioxidant activity of VPA was evaluated using the DPPH analysis method [42]. Compared with ascorbic acid, the antioxidant activity of VPA was found to be approximately 35%, exhibiting lower antioxidant capacity than ascorbic acid. Our study observed that NP3, which contains only VPA, exhibited low antioxidant activity. Our study used the DPPH and ABTS antioxidant analysis methods, which involved experiments where exogenous antioxidant substances such as BHT and ascorbic acid were taken as positive controls. The information in the literature and the antioxidant activity analyses in our study suggest that VPA does not exhibit strong antioxidant activity in the exogenous class, which could be attributed to its chemical structure. However, it should be noted that in the literature, the reparative effect of chitosan and VPA on oxidative stress is reported to manifest through good antioxidant activity in enzymatic reactions involving antioxidant enzyme systems (e.g., glutathione S-transferase (GST), glutathione reductase (GR), glutathione peroxidase (GPx), superoxide dismutase (SOD), and catalase (CAT)) or non-enzymatic reactions like lipid peroxidation [43,44]. However, NP4, which underwent dual drug loading through the addition of Que, showed antioxidant activity close to the BHT standard. Our study results are consistent with similar findings in the literature. The analysis results indicate that the antioxidant property of Que can be successfully utilized in NP systems, and this property of Que persists within the NP system.

The physicochemical properties of drug-loaded NPs are known to significantly affect cellular uptake and biological processes for these NPs to manifest their potential effects in cells. Among the key factors affecting the penetration of NPs into cells in the literature are particle size, particle shape, and charge [45,46]. Particles are absorbed through different endocytic pathways, each exhibiting distinct uptake dynamics. It has been reported that particles with diameters greater than 500 nm are internalized through phagocytosis, while particles with diameters smaller than 200 nm are generally internalized through pinocytosis [46]. Some studies have indicated that rod-shaped NPs exhibit higher cellular uptake than spherical ones due to their larger contact area with cell membrane receptors [46]. Additionally, positively charged chitosan NPs have been shown to adhere to negatively charged cell surfaces, leading to increased cellular uptake [45,46]. In our study, the synthesized NPs had an average particle size smaller than 200 nm, and Que-loaded NPs had a rod-like shape and were developed with positively charged chitosan. These characteristics suggest that the NPs can undergo cellular uptake processes. In this way, in our study, the antioxidant activity of the developed NPs was analyzed on H202-induced SH-SY5Y cells selected for evaluating neurotoxicity processes. H_2_O_2_ is commonly used as an inducer of oxidative stress in in vitro models. As a result of the MTT test, the IC_50_ value was determined as 200 µM (Figure 7). Additionally, the potential cytotoxic effects of free VPA, Que, chitosan, and NPs were determined after 24 h (Figure 8) and 48 h (Figure 9). Except for high doses of NP4, none of the samples showed significant cytotoxic effects at the concentrations used in this study (Figure 8 and Figure 9). At the concentration of 96 µg/mL applied to the cells, NP4 was observed to reduce cell viability to below 50%. When reviewing studies examining the cytotoxic effects of VPA, Que, and chitosan in SH-SY5Y cells, it was observed in a study by Suematsu et al. (2011) that Que did not demonstrate any significant cytotoxic effect in the range of 0–100 µM [47]. Manigandan et al. (2019) reported that chitosan did not show any significant cytotoxic effect at levels between 0 and 50 µM [48], while Terzioglu Bebitoglu et al. (2020) found that VPA at concentrations of 1 mM, 5 mM, and 10 mM did not exhibit any cytotoxic effect on SH-SY5Y cells [49]. In our study, antioxidant effects were explored in a model of H_2_O_2_-induced oxidative stress in SH-SY5Y cells. The mechanism of H_2_O_2_-induced cell damage includes the production of reactive hydroxyl radicals and byproducts of Fenton’s reaction that further interact directly with cellular components to damage proteins, lipids, and DNA [50]. Antioxidants can block these effects. Cell viability was measured 24 h and 48 h after H_2_O_2_-induced damage. Suematsu et al. (2011) examined the protective effects of Que against the death of H_2_O_2_-treated SH-SY5Y cells [47]. When SH-SY5Y cells were incubated in a medium containing 100 µM H_2_O_2_ and Que, H_2_O_2_-mediated damage was suppressed. When SH-SY5Y cells were cultured with 100 µM H_2_O_2_, cell viability decreased to approximately 38% that of the vehicle-treated cells. In the presence of Que, the viability of H_2_O_2_-treated cells increased in a Que-concentration-dependent manner, reaching about 67% of that of the vehicle-treated ones at 100 µM Que [47]. Xi et al. (2012) studied the protective effects of Que in H_2_O_2_-induced SH-SY5Y cells [51]. SH-SY5Y cells were pre-cultured with Que in different concentrations for 12 h or 10 µM for various periods, and then pre-conditioned cells were treated with H_2_O_2_ (0.5 mM, 12 h). The study revealed that Que arrested cellular damage. In the 2018 study by Han et al., the antioxidant effect of Que-loaded NP was analyzed in SH-SY5Y cells [52]. The dichlorodihydrofluorescein diacetate (DCFH-DA) fluorescent probe was selected to determine the total intracellular ROS level, and 200 µM H_2_O_2_ was used as a positive control in the study. Que powders could reduce amounts of ROS products to some extent, as indicated by most of the studies. When Que NPs were introduced, the ROS level was neutralized, similar to the cell control, indicating that Que NPs exhibit a unique ROS-scavenging activity. Cell viability increased from 40% to 70% with 10 µg/mL of Que NPs [52]. Similar to the literature, in our study, a neurotoxic environment was created by applying 200 µM H_2_O_2_ to SH-SY5Y cells (Figure 7). In our study, the antioxidant activity of free Que and NP2-containing Que became apparent starting at 8 µg/mL after 24 h (Figure 10). The highest efficacy was achieved by applying 96 µg/mL Que (*p* < 0.0005) and 48 µg/mL NP2 at 24 h, resulting in approximately 66.0% cell viability (*p* < 0.0001) (Figure 10). Additionally, it was observed that the antioxidant activity of free Que and Que-containing NP2 was initiated at 24 µg/mL for free Que and 48 µg/mL for NP2 after 48 h of application (Figure 11). The highest efficacy was achieved by applying 72 µg/mL Que (65.99% cell viability) and 96 µg/mL NP2 at 48 h, resulting in 77.30% cell viability (*p* < 0.0001) (Figure 11). Our study demonstrates a similarity to the literature regarding the antioxidant effect of Que on H_2_O_2_-induced SH-SY5Y cells.

All NPs developed in our study were synthesized using chitosan. Publications indicate that chitosan positively enhances cell viability in H_2_O_2_-induced SH-SY5Y cells. In a study conducted by Manigandan et al. (2019), applying 5–20 µM chitosan to H_2_O_2_-induced SH-SY5Y cells was reported to increase cell viability by approximately 70% [48]. Similarly, in our study, it was observed that both free chitosan and all NPs developed using chitosan increased cell viability in H_2_O_2_-induced cells. The application of 24 µg/mL, 48 µg/mL, 72 µg/mL, and 96 µg/mL chitosan significantly increased cell viability in a dose-dependent manner at both 24 h and 48 h. Within 24 h, cell viability increased between 52.54% and 61.04% in a dose-dependent manner (Figure 10), and within 48 h, cell viability increased between 62.12% and 66.31% in a dose-dependent manner (*p* < 0.0001) (Figure 10). The highest efficacy was achieved with the application of 96 µg/mL chitosan at 48 h, reaching 70% cell viability (*p* < 0.0001) (Figure 11). The application of NP1 (blank chitosan NP) showed a protective effect against H_2_O_2_ toxicity at concentrations of 72 µg/mL and 96 µg/mL (*p* < 0.0001). Applying 96 µg/mL NP1 for 24 h and 48 h raised cell viability to over 65%, with the highest efficacy observed at 48 h with 96 µg/mL NP1, resulting in 68% cell viability (*p* < 0.0001). Similar to the findings in the literature, the results of our study also indicate a positive effect of chitosan on cell viability. This suggests the potential for a synergistic effect in antioxidant activity within NPs developed with chitosan, owing to its protective impact.

In our study, although the in vitro DPPH and ABTS analyses showed low radical-scavenging activity for chitosan and NP1, positive effects on cell viability against hydrogen-peroxide-induced cell damage were observed for both chitosan and NP1. This indicates that the protective effect of chitosan may not be solely due to its direct scavenging effect on free radicals but could involve its relationship with the antioxidant enzyme system associated with oxidative stress. Furthermore, the strong antioxidant activity of Que observed in both in vitro DPPH and ABTS analyses, along with its ability to preserve cell viability in oxidatively stressed SH-SY5Y cells, suggests that NPs developed with Que, particularly in sensitive areas related to the brain, may have a therapeutic effect. Considering the protective effects of both chitosan and Que on cell viability, it is hypothesized that NPs developed with these two molecules may exert a synergistic effect, enhancing the potential for treatment against oxidative stress. In a study by El-Denshary et al., 2015, the synergistic protective role of chitosan NPs alone and combined with Que against oxidative stress and hepatotoxicity in rats was investigated [53]. The study applied 100 nm sized chitosan NPs alone and in combination with quercetin to rats in a toxic environment induced by carbon tetrachloride (CCl4). Que was not loaded into the NPs but was administered as a combined treatment. The study revealed that in the control group, the aspartate aminotransferase (AST) and alanine aminotransferase (ALT) enzyme values were 63.71 ± 1.48 and 67.29 ± 2.49, respectively, while in the CCl4-treated group, these values increased to 94.29 ± 2.15 and 212.57 ± 5.11, respectively. In the CCl4 group treated with chitosan NPs, AST and ALT enzyme values decreased to 79.50 ± 0.56 and 88.5 ± 3.80, respectively. In contrast, in the chitosan NP group treated with combined Que, these values further decreased to 73.56 ± 0.82 and 67.78 ± 3.50, respectively. The study also identified a synergistic effect of chitosan and Que for other oxidative stress parameters [53]. Similarly, in our study, it was observed that NPs containing chitosan and Que increased and protected cell viability against 50% cell damage induced by H_2_O_2_. While cell viability was 68% in cells with quercetin-containing NP2, cell viability was 77.30% in cells with empty chitosan NP1. Both NP1 and NP2, containing chitosan and Que molecules, exhibited antioxidant effects singly and synergistically, as reported in the literature.

In a study conducted by Terzioglu Bebitoglu et al. (2020), SH-SY5Y human neuroblastoma cells were pretreated with 1 mM, 5 mM, or 10 mM VPA and then compared to 15 mM glutamate exposure [49]. The MTT test was applied to determine cell viability, and the treatment with 1 mM VPA effectively increased the viability of cells exposed to glutamate for 24 h. In our study, the effect of free VPA as well as single- and dual-drug-loaded NPs on cell viability after H_2_O_2_ treatment was investigated. When cells were treated with 48 µg/mL (*p* < 0.01), 72 µg/mL (*p* < 0.01, *p* < 0.05), and 96 µg/mL (*p* < 0.0001, *p* < 0.01), an increase in cell viability was observed at 24 h and 48 h. Moreover, an evident increase in cell viability was observed with dose escalation. The highest efficacy was achieved by applying 96 µg/mL VPA at 24 h, reaching 57.89% cell viability (*p* < 0.0001) (Figure 10). However, no significant effect on cell viability was observed with low or medium doses of VPA (2 µg/mL, 8 µg/mL, and 24 µg/mL) (*p* > 0.05). The positive effect of VPA on cell viability in our study is consistent with the literature. In NP3 applications, cell viability remained above 60% in all dose applications at both time points (24 h and 48 h) (Figure 10 and Figure 11). NP3 exhibited the highest neuroprotective effect when applied at 2 µg/mL, maintaining cell viability up to 70%. While free VPA does not exhibit antioxidant activity at low doses, it is observed that NP3 (chitosan-loaded VPA NPs) shows antioxidant activity even at low doses. This suggests a synergistic effect of chitosan and VPA molecules on the NP3 content, contributing to antioxidant activity.

In the experiment with the 24 h application, low and medium doses of NP4 (2 µg/mL, 8 µg/mL, 24 µg/mL, and 48 µg/mL) significantly increased cell viability up to 66% (Figure 10). In the 24 h pretreatment experiment, NP4 provided lower protection against neurotoxicity at concentrations of 72 µg/mL and 96 µg/mL (Figure 10). In other words, the optimal dose for the best antioxidant effect of NP4 was determined to be 2 µg/mL at 24 h. In the 48 h pretreatment experiment, low, medium, and high doses (2 µg/mL, 8 µg/mL, 24 µg/mL, 48 µg/mL, 72 µg/mL, and 96 µg/mL) significantly increased cell viability between 54.13% and 60.78% (Figure 11). The efficiency of 48 µg/mL, 72 µg/mL, and 96 µg/mL NP4 was lower, whereas 2 µg/mL, 8 µg/mL, and 24 µg/mL NP4 protected cell viability by up to 60.78%, 60.63%, and 64.09% against H_2_O_2_, respectively. In our study, free VPA demonstrated antioxidant effects at high doses, while, when used in conjunction with chitosan and Que within NPs, it could exhibit antioxidant effects at lower doses. This indicates that combining bioactive molecule contents in nanoparticle formulations can achieve antioxidant effects at lower doses. However, the inability of NP4 to demonstrate antioxidant effects at high doses is thought to be related to the cytotoxicity analysis of NP4, where high doses may reduce the cell viability of healthy cells (Figure 8 and Figure 9).

This study’s findings demonstrate the in vitro antioxidant activities of the developed NPs and their protective effects in the H_2_O_2_-induced oxidative stress model. In vitro antioxidant capacity analyses have revealed that Que-containing NPs, especially NP2 and NP4, exhibit activity close to the antioxidant activity capacity of the BHT standard (Figure 6). While the antioxidant activity of NP1 and NP3 without Que was found to be lower compared with the antioxidant activity of BHT (Figure 6), all developed NPs were observed to protect against H_2_O_2_-induced cell damage and exhibit protective effects against oxidative stress in cell culture experiments (Figure 10 and Figure 11).

In cell culture experiments, the protective effects of NPs on SH-SY5Y cells were assessed. The NP applications that provided the highest protection against H_2_O_2_-induced cell damage (Figure 12) were determined: NP2 at 96 µg/mL (77.30%, 48 h), NP3 at 2 µg/mL (70.06%, 24 h), NP1 at 96 µg/mL (68.31%, 48 h), and NP4 at 2 µg/mL (66.03%, 24 h).

Our study emphasizes the positive effects of chitosan on antioxidant activity. NP1, which contains chitosan, demonstrated a protective effect, especially at high doses, against H_2_O_2_ toxicity. Furthermore, the low doses of NP3 and NP4, containing VPA exhibiting antioxidant activity in SH-SY5Y cells exposed to oxidative damage, suggest a potential novel therapeutic approach against oxidative injury during VPA treatment. Additionally, it was observed that NP2 and NP4, containing Que, exhibited a significant protective effect against oxidative stress induced by H_2_O_2_. Despite the cytotoxicity observed at high doses of NP4, its antioxidant effect persisted at low doses. This suggests that NP4 has notable protective potential at appropriate doses, while higher doses may adversely affect cell viability. When evaluating studies in the literature regarding the effective use of drug-loaded NPs in brain diseases [54] and considering our analysis results, it is believed that NPs developed by loading drugs into chitosan may potentially play a role in alleviating the oxidative stress associated with neurodegenerative conditions.

## 5. Conclusions

Our study demonstrates a successful paradigm in the development of drug-loaded nanoparticles, emphasizing a significant achievement in the homogeneous and regular distribution of drugs in nanoparticles. Chitosan nanoparticles were successfully developed with single and dual drug loadings possessing antioxidant activity.

From this research, we conclude the following:i.The capability of chitosan nanoparticles developed at the nanometer scale to sustain cell viability in SH-SY5Y cell lines implies the potential of the intranasal administration of chitosan nanoparticles for future studies, offering protective effects in central nervous system diseases.ii.The antioxidant effect of chitosan is believed to be based on its ability to reduce intracellular oxidative stress mechanisms rather than directly inhibiting free radicals, as indicated by our cellular oxidative stress analyses.iii.The facile and strong binding ability of the flavonoid quercetin, which possesses antioxidant activity, to chitosan demonstrates the potential synergistic effect of quercetin-loaded nanoparticles with chitosan as a protective agent in the nervous system.iv.The safety of low doses of VPA and its protective effect in SH-SY5Y cells up to a certain threshold are emphasized.v.The anticipation that quercetin’s early release in dual-drug-loaded chitosan nanoparticles, alongside the controlled and slower release of VPA, could initiate antioxidant effects and mitigate dose-dependent VPA toxicity, represents significant insights for future in vitro and in vivo cellular and molecular analysis studies.

Nanoparticles developed through the combination of quercetin, VPA, and chitosan, especially at optimized doses, emerge as promising therapeutic agents for alleviating the oxidative stress associated with neurodegenerative conditions. These findings underscore the significance of single and dual drug loadings in nanotechnology, providing new and effective strategies for reducing oxidative stress during the pharmacological treatment of neurological diseases.

## Figures and Tables

**Figure 1 biomedicines-12-00287-f001:**
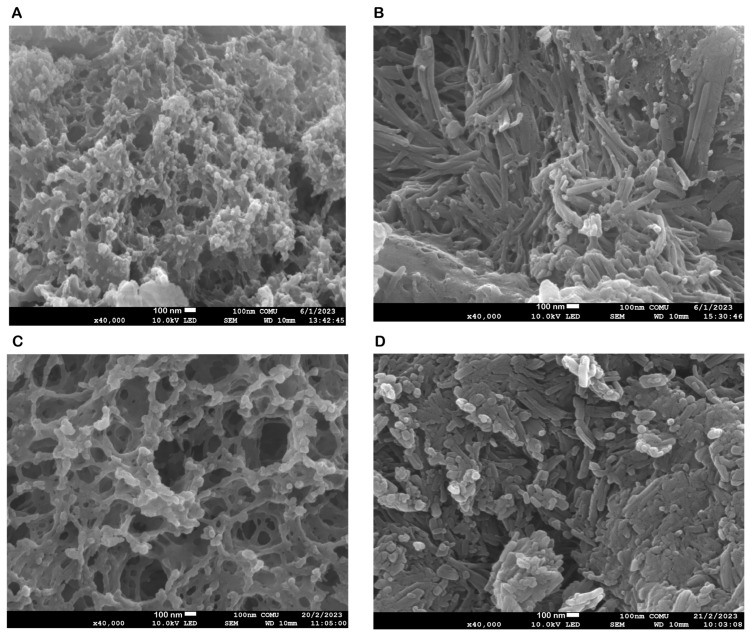
Field emission scanning electron microscopy (FE-SEM) morphological evaluation of the nanoparticles (NPs). (**A**) FE-SEM image of blank chitosan NP (NP1), (**B**) FE-SEM image of Que-loaded chitosan NP (NP2), (**C**) FE-SEM image of VPA-loaded chitosan NP (NP3), (**D**) FE-SEM image of Que- and VPA-loaded chitosan NP (NP4).

**Figure 2 biomedicines-12-00287-f002:**
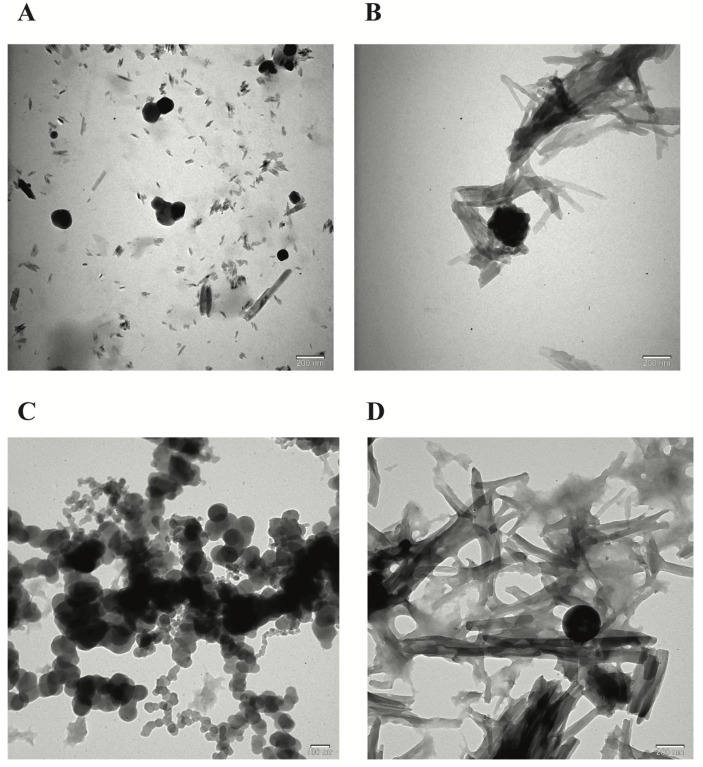
Transmission electron microscopy (TEM) morphological evaluation of the nanoparticles (NPs). (**A**) TEM image of blank chitosan NP (NP1), (**B**) TEM image of Que-loaded chitosan NP (NP2), (**C**) TEM image of VPA-loaded chitosan NP (NP3), (**D**) TEM image of Que- and VPA-loaded chitosan NP (NP4).

**Figure 3 biomedicines-12-00287-f003:**
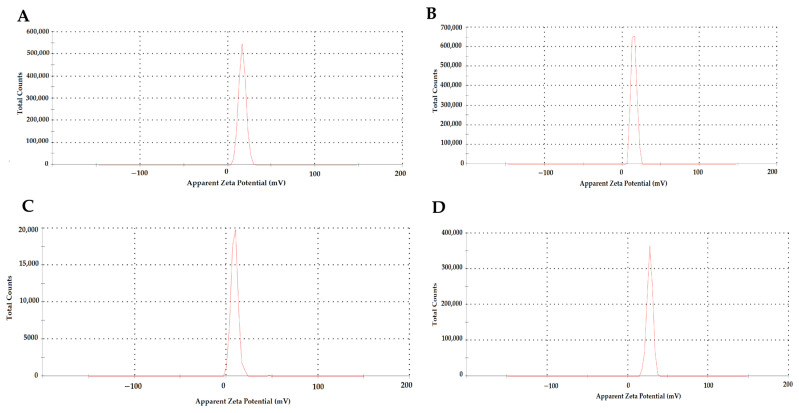
Zeta results of the developed nanoparticles (NPs) in the study. (**A**) NP1—blank chitosan NP; (**B**) NP2—Que-loaded chitosan NP; (**C**) NP3—VPA-loaded chitosan NP; (**D**) NP4—Que- and VPA-loaded chitosan NP.

**Figure 4 biomedicines-12-00287-f004:**
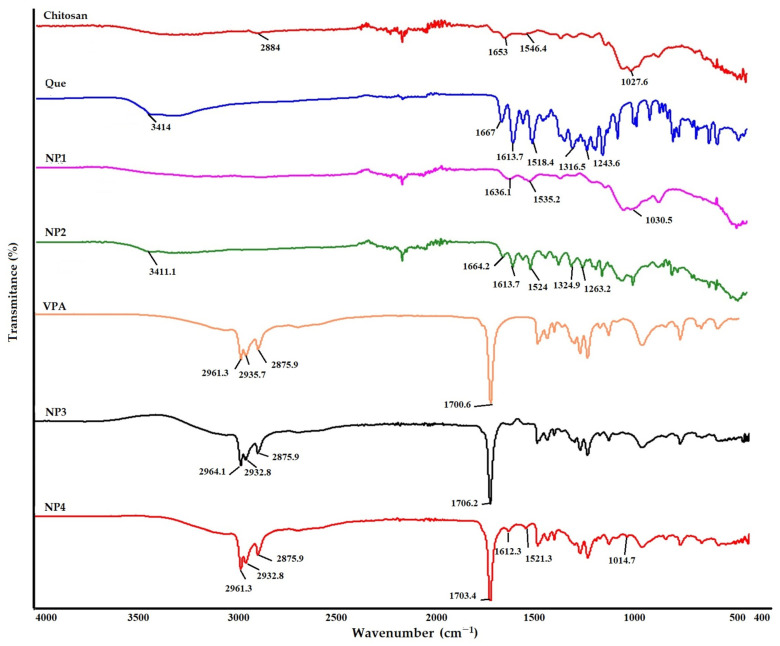
Fourier transform infrared spectroscopy (FT-IR) analysis results. FT-IR analysis results of chitosan, Que (Quercetin), NP1—blank chitosan nanoparticle (NP); NP2—Que-loaded chitosan NP; NP3—VPA-loaded chitosan NP; NP4—Que- and VPA-loaded chitosan NP, VPA (Valproic Acid).

**Figure 5 biomedicines-12-00287-f005:**
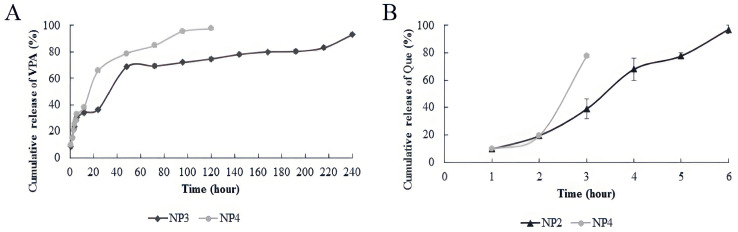
In vitro release analysis results. (**A**) In vitro release analysis results of VPA (Valproic Acid); (**B**) in vitro release analysis results of Que (Quercetin).

**Figure 6 biomedicines-12-00287-f006:**
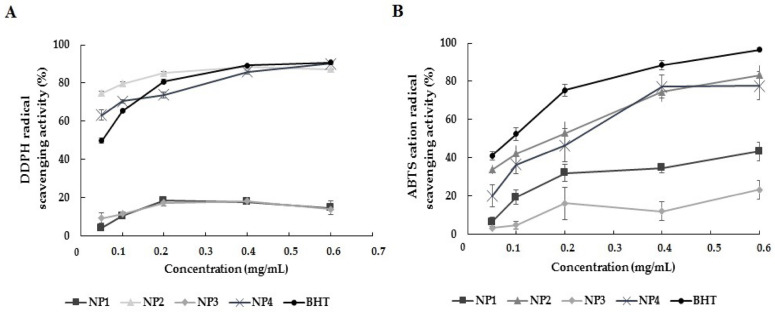
Results of in vitro antioxidant activity of the nanoparticles (NPs). NP1—blank chitosan NP; NP2—Que-loaded chitosan NP; NP3—VPA-loaded chitosan NP; NP4—Que- and VPA-loaded chitosan NP; Butylated hydroxytoluene (BHT) (**A**) DPPH activity analysis; (**B**) ABTS activity analysis.

**Figure 7 biomedicines-12-00287-f007:**
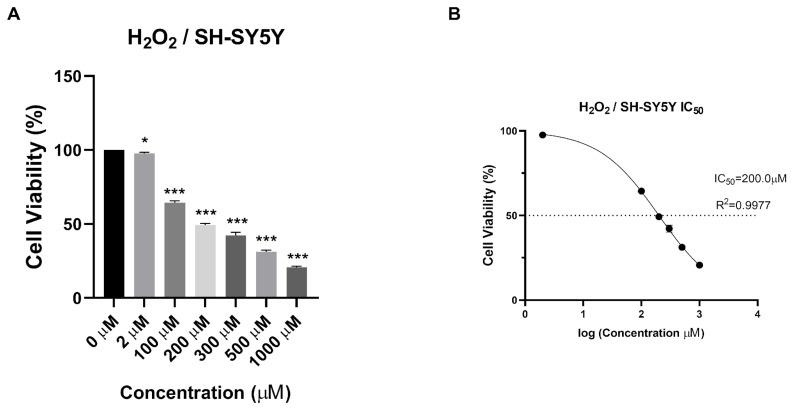
The cytotoxic effect of H_2_O_2_ with different concentrations (0 μg/mL, 2 μg/mL, 100 μg/mL, 200 μg/mL, 300 μg/mL, 500 μg/mL, and 1000 μg/mL) on SH-SY5Y cells at 24 h. (**A**) % cell viability of SH-SY5Y cells treated with H_2_O_2_ at different concentrations for 24 h. (**B**) The IC_50_ values of H_2_O_2_ for 24 h. Experiments were carried out in triplicate. Data are expressed as mean ± standard deviation (S.D.)., *** *p* < 0.0005, * *p* < 0.05.

**Figure 8 biomedicines-12-00287-f008:**
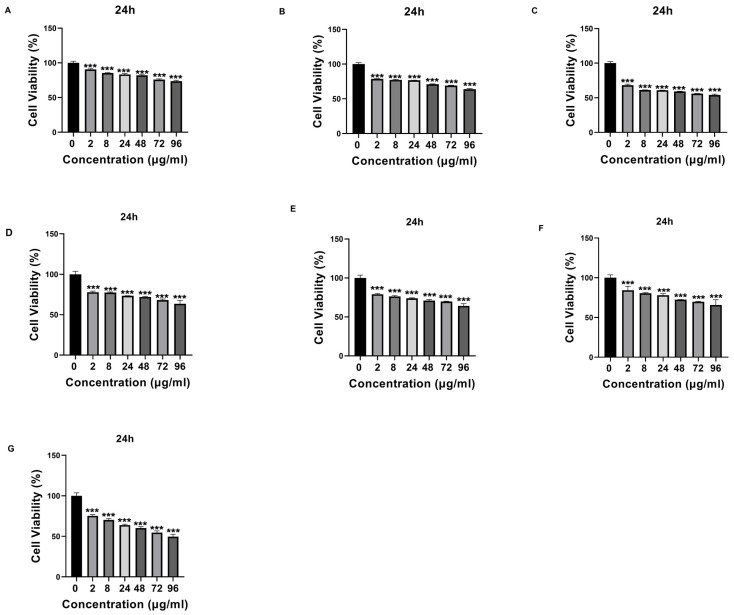
The cytotoxic effect of samples with different concentrations (0 μg/mL, 2 μg/mL, 8 μg/mL, 24 μg/mL, 48 μg/mL, 72 μg/mL, and 96 μg/mL) on SH-SY5Y cells at 24 h. (**A**) Valproic Acid (VPA), (**B**) Quercetin (Que), (**C**) Chitosan, (**D**) NP1—blank chitosan nanoparticle (NP), (**E**) NP2—Que-loaded chitosan NP, (**F**) NP3—VPA-loaded chitosan NP, (**G**) NP4—Que- and VPA-loaded chitosan NP. Experiments were carried out in triplicate. Data are expressed as mean ± standard deviation (S.D.). *** *p* < 0.0005.

**Figure 9 biomedicines-12-00287-f009:**
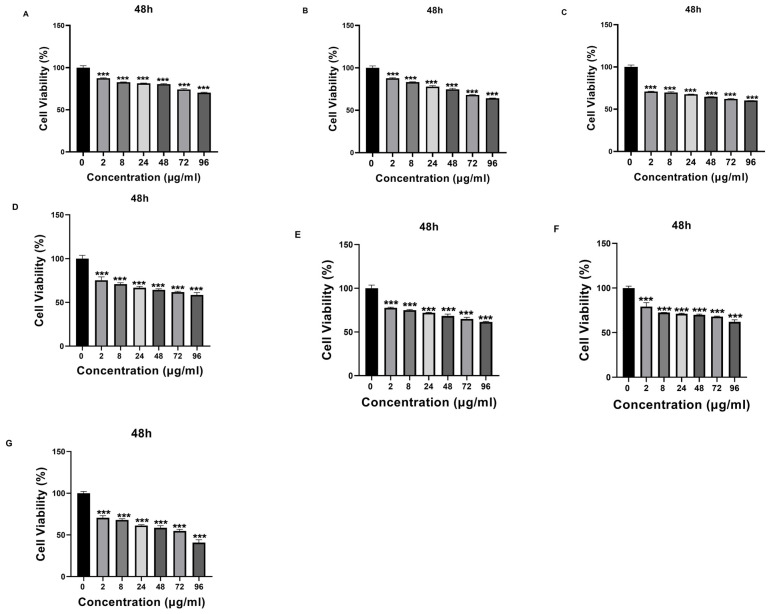
The cytotoxic effect of samples with different concentrations (0 μg/mL, 2 μg/mL, 8 μg/mL, 24 μg/mL, 48 μg/mL, 72 μg/mL, and 96 μg/mL) on SH-SY5Y cells at 48 h. (**A**) Valproic Acid (VPA), (**B**) Quercetin (Que), (**C**) Chitosan, (**D**) NP1—blank chitosan nanoparticle (NP), (**E**) NP2—Que-loaded chitosan NP, (**F**) NP3—VPA-loaded chitosan NP, (**G**) NP4—Que- and VPA-loaded chitosan NP. Experiments were carried out in triplicate. Data are expressed as mean ± standard deviation (S.D.). *** *p* < 0.0005.

**Figure 10 biomedicines-12-00287-f010:**
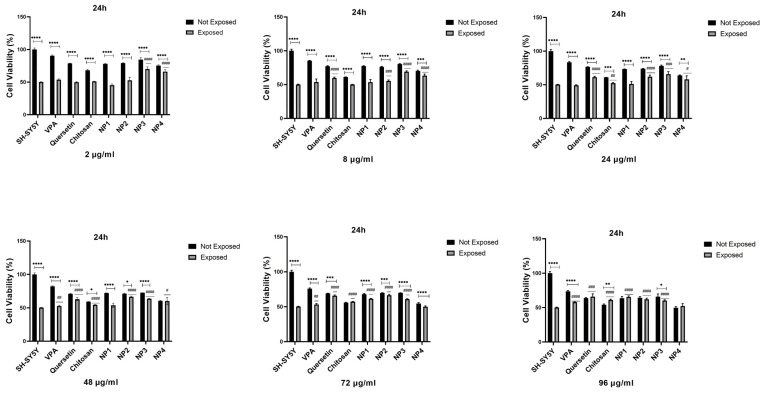
The antioxidant effect of samples with different concentrations (2 μg/mL, 8 μg/mL, 24 μg/mL, 48 μg/mL, 72 μg/mL, and 96 μg/mL) on H_2_O_2_-challenged SH-SY5Y cells for 24 h. VPA (Valproic Acid); NP1—blank chitosan nanoparticle (NP); NP2—Que-loaded chitosan NP; NP3—VPA-loaded chitosan NP; NP4—Que- and VPA-loaded chitosan NP. Experiments were carried out in triplicate. Data are expressed as mean ± standard deviation (S.D.). Compared to “exposed to H_2_O_2_” groups vs. “not exposed to H_2_O_2_” groups ****; *p* < 0.0001, ***; *p* < 0.0005, **; *p* < 0.01, *; *p* < 0.05. Compared to “exposed to H_2_O_2_“ on SH-SY5Y group vs. “exposed to H_2_O_2_^”^ sample groups ####; *p* < 0.0001, ###; *p* < 0.0005, ##; *p* < 0.01, #; *p* < 0.05.

**Figure 11 biomedicines-12-00287-f011:**
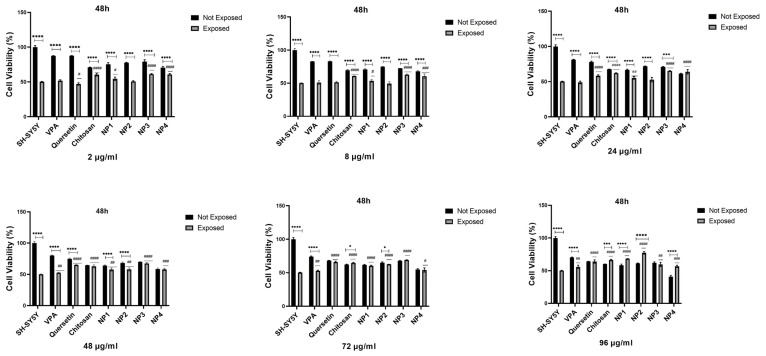
The antioxidant effect of samples with different concentrations (2 μg/mL, 8 μg/mL, 24 μg/mL, 48 μg/mL, 72 μg/mL, and 96 μg/mL) on H_2_O_2_-challenged SH-SY5Y Cells for 48 h. VPA (Valproic Acid); NP1—blank chitosan nanoparticle (NP); NP2—Que-loaded chitosan NP; NP3—VPA-loaded chitosan NP; NP4—Que- and VPA-loaded chitosan NP. Experiments were carried out in triplicate. Data are expressed as mean ± standard deviation (S.D.). Compared to “exposed to H_2_O_2_“ groups vs. “not exposed to H_2_O_2_” groups ****; *p* < 0.0001, ***; *p* < 0.0005; *; *p* < 0.05. Compared to “exposed to H_2_O_2_” on SH-SY5Y group vs. “exposed to H_2_O_2_” sample groups ####; *p* < 0.0001, ###; *p* < 0.0005, ##; *p* < 0.01, #; *p* < 0.05.

**Figure 12 biomedicines-12-00287-f012:**
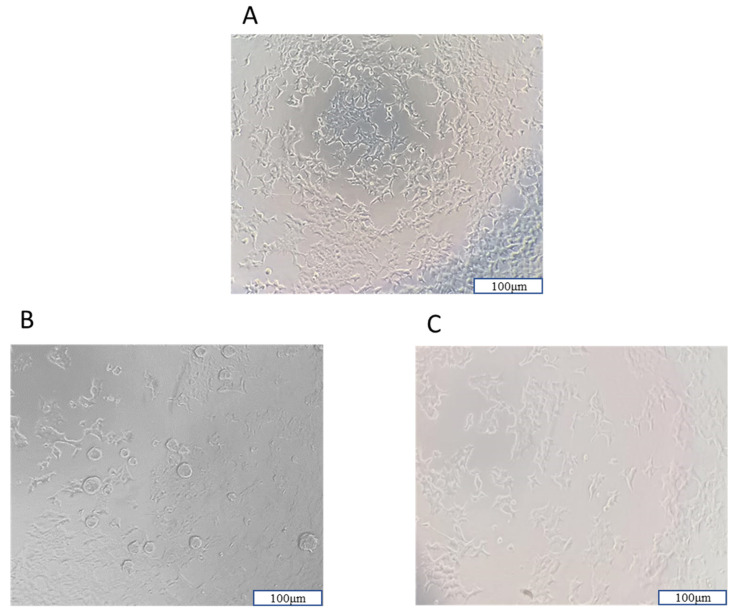
Images of cell viability of SH-SY5Y cells are included in this study. (**A**) Untreated SH-SY5Y, (**B**) H_2_O_2_-induced damage in the SH-SY5Y cell, (**C**) NP exposure to H_2_O_2_-induced damage in the SH-SY5Y cell. Scale bar = 100 µm with the use of a 10× magnification fluorescence microscope (Olympus CK40M).

**Table 1 biomedicines-12-00287-t001:** Results of characterization analysis of nanoparticles.

Nanoparticle	Particle Size (nm) (Mean ± Std. Dev.)	Zeta Potential (mV) (Mean ± Std. Dev.)	PDI (Mean ± Std. Dev.)	Encapsulation Efficiency (%) (Mean ± Std. Dev.)	Drug Loading (%) (Mean ± Std. Dev.)
NP1	153.63± 12.81	16.60 ± 4.12	0.08 ± 0.12	NC	NC
NP2	180.52 ± 8.23	15.10 ± 3.42	0.05 ± 0.47	93.46 ± 5.52	62.22 ± 7.73
NP3	93.21 ± 7.25	8.89 ± 3.82	0.08 ± 0.73	99.9 ± 7.88	37.71 ± 6.52
NP4	198.73 ± 9.13	27.20 ± 3.67	0.05 ± 0.94	99.23 ± 7.81 (for Que) 97.8 ± 5.42 (for VPA)	39.16 ± 7.23 (for Que) 25.75 ± 5.98 (for VPA) 64.91 ± 5.57 (for total dual drug)

NP1—blank chitosan NP; NP2—Que-loaded chitosan NP; NP3—VPA-loaded chitosan NP; NP4—Que- and VPA-loaded chitosan NP; PDI—polydispersity index; NC—not calculated; Que (Quercetin); VPA (Valproic Acid).

## Data Availability

Data are available upon request from the corresponding author.

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
