# Peer review of "Chitosan Nanoparticles Loaded with Quercetin and Valproic Acid: A Novel Approach for Enhancing Antioxidant Activity against Oxidative Stress in the SH-SY5Y Human Neuroblastoma Cell Line"

_biomedicines, 2024, doi:10.3390/biomedicines12020287_

Round 1
Reviewer 1 Report
Comments and Suggestions for Authors
In this manuscript, the authors successfully developed chitosan NPs loaded with both single and dual drugs with antioxidant activity. The NPs were prepared through the mixture of the drugs and chitosan, which may serve as therapeutic agents to alleviate the oxidative stress associated with neurodegenerative diseases. However, the work is lack of scientific significance and novelty, and the majority of the results were not well-explained. Therefore, I do not recommend publication of this manuscript on Biomedicines. Before the submission to another specialized journal, please the authors address the following issues:
1. What are the advantages to combine two antioxidants in a single nanoparticle? Could the combinational treatment improve the therapeutic results. If so, what is the mechanism?
2. From the TEM images in Figure 2, it can be observed that the chitosan NPs aggregated after the loading of the drug. The morphology for the quercetin and valproic acid-loaded NPs changed to rods and spheres, respectively. The reason for the changes should be explained.
3. The aggregated NPs will surely influence their cellular uptake. When the size of the NPs is larger than 1 µm, the endocytosis is greatly inhibited. How does the drug loaded in the Que and VPA-loaded chitosan NPs get into the cells?
4. In Figure 5, the NP3, which is the VPA-loaded chitosan NP, did not exhibited any antioxidative activities. Why?
5. In the cytotoxicity experiments, a concentration of 96 µg/mL was applied to the cells. The NP4 exhibited a certain cytotoxicity at the concentration, which killed >50% of the cells. So the antioxidative experiment of this nanoparticle is meaningless.
6. The Que and VPA did not demonstrate any synergistic effect.
7. Is there other evidence to prove the antioxidative effect of the NPs on cellular level, such as the expression of proteins and other biomarkers?
8. Please provide the loading ratio of the drugs in the NPs.
9. Why is the release curve of the drugs inconsistent in Figure 5C? So is Figure 5D.
10. Some figures are not clear, such as Figure 7 and 8, and some words in the figures are too small, such as Figure 9 and 10.
11. The position of the Figures should be consistent throughout the entire text.
Comments on the Quality of English LanguageThere are many errors in the manuscript, such as:
1. Please revise the “ml” to “mL”.
2. The 3 in the “5×103” should be superscript.
3. The 2 in the “H2O2” should be subscript.
Author Response
Dear Reviewer,
I want to express my gratitude for your valuable evaluation. Our paper can be further enriched with your insightful critiques. Taking into consideration your contributions, we have revised our work. Please consider our revisions throughout the process.
As you pointed out, to emphasize the scientific importance and novelty of our paper, we have updated our literature review and added more scientific context to enhance the overall value of our study.
During the revision process, we have delved into the Methodology, Findings, and Discussion sections, providing more detailed explanations, addressing potential shortcomings, and striving to make our results more comprehensible. All changes made within the article have been highlighted in red.
In line with your suggestions, we have focused on the areas you highlighted and incorporated changes for each recommendation and question within your feedback. We are committed to making our paper stronger with your valuable suggestions. Thank you for your interest and support.
Best regards,

Reviewer 2 Report
Comments and Suggestions for Authors
This manuscript aims to develop a novel nanoparticle (NPs) by loading Quercetin (Que) and valproic acid (VPA) onto chitosan. The authors synthesized chitosan NPs loaded with either single (Que or VPA) or dual drugs (Que and VPA), characterized the NPs, conducted in vitro antioxidant activity studies, and analyzed the cytotoxicity and antioxidant activity of the NPs in SH-SY5Y cell lines. A paradigm for the development of drug-loaded NPs was successfully established, ensuring the uniform and homogeneous distribution of drugs on the NPs. Antioxidative chitosan NPs were successfully developed, loaded with both single and dual drugs. This approach holds promising prospects for achieving synergistic effects between drugs or overcoming undesired effects. However, there are also issues that need to be addressed:
Question 1:In Figure 1, please incorporate a scale bar to indicate the image scale. Simultaneously, ensure that the information in the figure is adjusted for greater clarity and accuracy.
Question 2:Ensure that the data in Table 1 is accurate to two decimal places to enhance the precision and credibility of the data.
Question 3:In line 272 on page 8, the expression "....polydispersity index, -; not detected." lacks clarity in symbol representation. Please provide an explanation and clarification.
Question 4:In the abstract, it is suggested to include a statement indicating potential future directions or applications based on the research results, providing continuity beyond the scope of the current study.
Question 5:In Figure 4A, the peaks indicated in the Fourier-transform infrared spectrum, such as "3313, 3245, 2864," are not visually apparent. It is recommended to adjust the graphics or provide a more detailed explanation.
Question 6:In Figures 4C and 4D, no error bars are present. Please explain or supplement relevant information to enhance the completeness of the figures.
Question 7: Please adjust the clarity of Figures 9 and 10 to ensure the quality and sharpness of the images.
Question 8:In Figure 11, please specify the magnification factor and scale to provide additional information.
Question 9:The manuscript extensively elaborates on the rationality of utilizing nanoparticle (NP) formulations, considering the limitations of VPA and Quercetin (Que). To enhance clarity, a brief mention could be made of how NPs potentially address the mentioned limitations, such as improving bioavailability and reducing toxicity.
Question 10:The justification for choosing chitosan as a material for NP development is well-founded. However, a brief elaboration on the unique properties of chitosan, emphasizing its biocompatibility, biodegradability, and FDA approval, could be included.
Question 11:Although the manuscript discusses the release patterns of Que and VPA from NPs at pH 7.4, the importance of the observed release patterns lacks discussion. How do the differences in release times between NP3 and NP4 potentially impact the applications of these nanoparticles? It is suggested to provide a more detailed exploration of this aspect.
Comments on the Quality of English LanguageEnglish language of the manuscript needs minor editing.
Author Response
Dear Reviewer,
I want to express my gratitude for your valuable evaluation. Your insightful critiques can further enrich our paper. We have revised our work, taking into consideration your contributions. Please consider our revisions throughout the process.
As you pointed out, to emphasize the scientific importance and novelty of our paper, we have updated our literature review and added more scientific context to enhance the overall value of our study.
During the revision process, we have delved into the Methodology, Findings, and Discussion sections, providing more detailed explanations, addressing potential shortcomings, and striving to make our results more comprehensible. All changes made within the article have been highlighted in red.
In line with your suggestions, we have focused on the areas you highlighted and incorporated changes for each recommendation and question within your feedback. We are committed to making our paper stronger with your valuable suggestions. Thank you for your interest and support.
Best regards,

Reviewer 3 Report
Comments and Suggestions for Authors
How the optimal time was determined for stirring the samples (24 hours) determined? Random, according to the literature, or through trials? Ditto for the centrifugation time.
The efficiency of NPs is dependent on their size and dimensional distribution, which must be within a determinedrange of dimensions. I recommend, if possible, a study by dynamic light scattering (DLS) that could complete the discussions regarding the dimensional analysis.
I request the authors, in the conclusions, they present synthetically the most relevant results of their study, i.e. those results that can convince a potential user to finance the development of an in-depth study with transferable results.
Also an arrangement of the results in order of quality, so as to easily recognize the increase in efficiency depending on NP chitosan, NP loaded with Que, NP loaded with VPA and NP loaded with VPA and Que.
Author Response

(The authors gave the same response as above.)

Round 2
Reviewer 1 Report
Comments and Suggestions for Authors
The authors have answered the questions from the reviewers. Although there are still concern regarding to the size of the nanoparticles, the quality of the manuscript has been improved. I agree publication of this version on Biomedicines.
Comments on the Quality of English LanguageEnglish is good enough for publication.